# An Online Experiment of NHS Information Framing on Mothers’ Vaccination Intention of Children against COVID-19

**DOI:** 10.3390/vaccines10050720

**Published:** 2022-05-04

**Authors:** Audrey L. Van Hoecke, Jet G. Sanders

**Affiliations:** Department of Psychological and Behavioural Science, The London School of Economics and Political Science, London WC2A 3PH, UK; j.g.sanders@lse.ac.uk

**Keywords:** COVID-19, flu, children, mothers, parents, vaccination intention, vaccination attitude, risk framing, vaccination hesitancy

## Abstract

Children under the age of 5, will likely all be offered vaccination against SARS-CoV-2 soon. Parental concerns over vaccination of children are long standing and could impede the success of a vaccination campaign. In the UK, a trusted source to inform vaccination choices is the NHS website. Here we used a randomized controlled experiment of framing effects in NHS information content for COVID-19 and flu with 550 mothers under the age of 5. We compared both vaccination offers following two commonly used frames in vaccination informational campaigns: alerting to the risks of no vaccination for the child itself vs. those in their community. We find that vaccination intention was twice as high when risks to the child are emphasized, relative to risks to the community. Exploratory analyses suggest that these effects may differ between white and non-white mothers. Whilst communication directed at adult vaccination against COVID-19 generally focuses on risks of infecting others, communication about vaccination of children may benefit from emphasizing risks to the children themselves. This pattern is in line with flu vaccination research from pre-COVID-19 times.

## 1. Introduction

As of February 2022, more than 5.9 million people have died from COVID-19 [1]. According to UNICEF, 4 out of 1000 COVID-19 deaths were children [2]. With the adult population being immunized and preventive measures coming to an end, uptake of vaccines among young children may be crucial both to protect them from the disease and, indirectly, to protect the rest of the population [3]. Therefore, understanding the factors that drive and affect parental intention to vaccinate their children is essential. This is important in particular for mothers, as they are likely to spend more time with their under 5 years old children (78% more than men), especially during the pandemic [4].

Three main categories of factors play a significant role in maternal vaccine hesitancy (1) external factors, including fake news and conspiracy theories, which have been central to research [5,6,7], (2) increased urge to be involved in their children’s healthcare decisions [8] and (3) idiosyncratic factors such as the parental cognitive strain, lack of numerical literacy and cognitive biases [7,8,9,10]. Most of this literature is based on recommended childhood vaccines, mainly the influenza vaccine [11].

This evidence shows that parents struggle to make sense of the abundance of information available on vaccination, often leading them to make predictable mistakes about vaccination [12,13,14]. Therefore, campaigns often aim at simplifying information and providing parents with a clear take-home message coming from familiar and trusted sources. Studies have shown that such campaigns positively impact the parental decision process [15,16].

Prior to the pandemic, one common strategy to formulate a campaign was disputing the perceived dangers of immunization. This approach often turned out ineffective for two main reasons: first, following principles of confirmation bias, vaccine hesitant individuals are unlikely to make a swift transition to contradictory beliefs [15,17,18]. Second, offering undeniable proof for the absence of risk can prove difficult [15,19,20].

For this reason, recent behavioral strategies in childhood vaccination communication have focused on providing a unified frame, replacing an existing fear (e.g., vaccination side effects) with an alternative one (e.g., illness), as opposed to countering the existing fear [9,19,20,21,22]. Alternative frames often focus on risks to the individual, and sometimes on risks to the immediate community or society.

A frequent object of debate is whether to describe risks to the individual or community. Several studies prior to and during COVID-19 have been found to increase vaccination intention when vaccination is framed as a public good that helps create a layer of protection for vulnerable groups [23,24,25,26,27,28,29]. With regards to parental intentions as well, ‘benefit to others’ has been established as an important motivating factor [30].

The frame also presents one major drawback: benefits to others highlight that individuals could profit from free-riding in a well-vaccinated community, without contributing to the collective effort [30]. This drawback can be particularly visible under circumstances where a perceived vulnerable group (e.g., children) is asked to support the community. Unsurprisingly then, this frame has also led to decreased children immunization intentions [31].

Similarly, there are limitations to using an individual level risk frame for invitations to vaccination of children. This frame is more commonly used for diseases with severe consequences for children [32]. However, both COVID-19 and the flu present relatively low (flu) or very low (COVID-19) risks for children. This may make the personal benefit less clear and the frame less appropriate for these types of disease. 

With this in mind, we set out to test the effectiveness of these two frames on a frequently visited informational source for parents, the NHS website. In the experiment we compared vaccination intention and vaccination attitude for parents viewing the vaccination information webpage. The information was framed either by the risks that not vaccinating represent for the children themselves or by alerting to risks for others in the community [23] for two vaccine-preventable respiratory and airborne diseases of relevance to the current times: the common flu and COVID-19. This to (1) verify whether transferability of results exists between different types of disease, and (2) to corroborate results with those prior to the COVID-19 pandemic [5,6,33].

To validate our sample, we will start by looking at the effects of person characteristics on vaccination intention and attitude. We predicted that we would (i) observe differences between the two psychological frames on the vaccination attitude and the mothers’ intention to vaccinate their children for the two disease types. In particular, (ii) we expected that the ‘community frame’ would be more effective the lesser the threat to the child, posing that the child frame would be more effective in a flu context while the community frame would be more effective in a COVID-19 context. By means of exploratory analyses, we will review treatment effects by person characteristics which differently impact vaccination intention and attitude in our sample. 

## 2. Methods

### 2.1. Participants and Sample

To recruit our sample, we used Prolific Academic [34] an online crowdsourcing platform that has been proven suitable in recruiting participants for social science experiments and has been validated for use of surveys experiments of general populations and marginalized groups [35]. The platform aims to reach out to as broad a sample of qualifying participants as possible and engages with its participants and researchers to cultivate a constructive research environment [36]. For this study, the criteria for selecting participants were the following: (i) being a mother above 18 years old, (ii) living in the UK, and (iii) having at least one child under 5 years old. The survey was created using Qualtrics and took 6 min to complete on average. Participants were offered financial compensation of £1.13 to ensure completion rates.

Sample size calculations (based on 80% power, an alpha of 0.05 and sample to accommodate small to medium effect size [30,36]), estimated 550 participants using G*power. To accommodate participation (i) completion rates, (ii) exclusions post allocation and (iii) failing of attention questions, we established that at least 91 participants were required per group to achieve the point of stability [37,38,39].

### 2.2. Design

We ran a 2 × 2 between-subjects experiment, comparing information about two types of airborne infectious diseases (Flu or COVID-19, IV1) by two types of risk frames (risks to the child or the community, IV2) on mothers’ intention to vaccinate their child(ren) under the age of 5, DV). Participants were randomly assigned to one of the four conditions (See Figure 1).

### 2.3. Materials

We created National Health Services (NHS)-like online informative pages on the benefits of vaccination against the flu and against COVID-19. The NHS is deemed a likeable and honest messenger for health-related information in the UK [4,16,40]. A webpage on vaccinating children against the flu was adapted to (i) minimize the duration of the experiment and (ii) increase the comparability between COVID-19 and Flu information. 

Across conditions, articles appeared to be written by the NHS (see Figure 2) and included (i) a headline on the role of parents (ii) statistics about the number of deaths caused by the specific disease in the UK (annual figure for the Flu and the absolute figure for COVID-19) and (iii) a picture and text relevant to the specific psychological frame used. The text for the Flu condition was: “*Flu—or Influenza—is a very common, highly infectious disease, caused by a virus. The worry is that certain groups of people—the elderly, pregnant women, children under five and those with long-term health conditions—are at greater risk of becoming seriously ill from flu. And while many healthy people can fight off the flu, there can be complications leading to death. It is estimated that around 11,000 in England and Wales are attributed to influenza infections annually. During the 2017–2018 season that number increased to 22,000*”.

Under the Flu condition, the two psychological frame conditions varied in the following ways:

Child frame: “*Parents should worry about the flu this winter [headline]. Children under the age of 5 face risk of extreme illness or death. If your child is under the age of 5, they are in the high-risk group for flu and can develop deadly flu-related complications, most commonly a bacterial chest infection, which can develop in pneumonia. Not vaccinating your child puts them at risk of life-threatening complications they can get from flu including meningitis and septic shock. It has been estimated that in the United States, flu-related hospitalizations among children younger than 5 years old can go up to 26,000 every year.*”

Community frame: “*As members of a community, parents need to worry about the flu this winter [headline]. If you don’t get your child vaccinated, you put your community at risk. Children are ‘super-spreaders’ who can pass the deadly virus to their grand-parents or other vulnerable people. This can have serious consequences on their health and can even lead to their death. Not vaccinating your child puts your entire community at risk, especially those who are at extreme risk of having deadly flu illness, like babies and young children, older people, healthcare professionals and people with certain long term health problems*.”

The text for the COVID-19 condition was: “*COVID-19 is a highly infectious disease, caused by a virus. The worry is that certain groups of people, the elderly, pregnant women, healthcare staff and those with long-term health conditions—are at greater risk of becoming seriously ill from COVID-19. And while many healthy people can fight off COVID-19, there can be complications leading to death. It is estimated that more than 44,000 deaths in the UK are attributed to COVID-19 since March 2020.*” It is worth noting that information provided under this condition, was based on the UK’s first coronavirus variant, Alpha.

Child frame: “*Parents should worry about COVID-19 this winter [headline]. Young children face the risk of hospitalization. Most reported infections in children are asymptomatic or mild. Less is known about severe COVID-19 cases in children leading to hospitalization. Analysis of COVID-19 hospitalization data in the US found that one in three hospitalized children was admitted to an intensive care unit. Not vaccinating your child, once a vaccine becomes available, puts them at risk of serious hospitalization and life-threatening complications. The hospitalization rate among children from early March 2020 to the end of July 2020 was 8 cases per 100,000, with the highest rate among children under 2 years old.*”

Community frame: “*As members of a community, parents need to worry about COVID-19 this winter [headline]. If you don’t get your child vaccinated, you put your community at risk. Children are ‘super-spreaders’ who can pass the deadly virus to their grand-parents or other vulnerable people. This can have serious consequences on their health and can even lead to their death. Not vaccinating your child, once a vaccine becomes available, puts your entire community at risk, especially those who are at extreme risk of having deadly COVID-19 illness, like older people, healthcare professionals and people with certain long term health problems.*”

To facilitate the salience [41] of the differential risk frames and disease context, we used a vivid image in all 4 contexts [17,42,43] (See Figure 2).

### 2.4. Measures

#### 2.4.1. Intention to Vaccinate

To assess vaccination intention, items by Myers & Goodwin, 2011; Quinn, Parmer, 2013, were adapted [44,45]. Participants were asked “Will you vaccinate your child (ren) against the flu/influenza this year?” in the Flu condition. Participants could select responses “Yes, I would like to vaccinate all my children (1) or some of my children (2) this year”, “Yes, I have already vaccinated some or all of my children this year”(3), “No, I will not vaccinate any of my children this year by choice”(4), “I am still undecided”(5), “I refuse to answer”(6).

For the analysis, the first (1–3) and latter three (4–6) response options were joined to create a binary outcome variable (Yes and No/Don’t know), as done by Shmueli in 2021 [46].

As the COVID-19 vaccination was yet to become available at the time of the survey, participants were asked: “Would you want to vaccinate your child against COVID-19, should a vaccine be made available to them?” This question was answered on a five-point Likert scale from “Most definitely not” (1), “Probably not” (2) “Don’t know yet/Unsure” (3) “Probably” (4) “Most definitely” (5). Here too, the first three (1–3) and latter two (4,5) response options were joined to create a binary outcome variable (Yes and No/Don’t know) for analysis.

In both conditions, participants also had the option to select “No, my child is medically exempt” (Flu), “My child would probably be medically exempt” (COVID-19). These participants were excluded from further analysis (See Figure 1).

#### 2.4.2. Attitude towards Vaccination

To assess attitude toward vaccination we used two common scales. The Vaccine Confidence Index (VCI) [47,48] measures the participant’s perception of three key aspects of vaccine uptake shown to predict influenza vaccination amongst adults and children [16,48,49] and COVID-19 vaccination amongst adults [50]: importance (“Vaccines are important for children to have”), safety (“Vaccines are safe”), and effectiveness (“Vaccines are effective”) of vaccines. Responses are coded on a 7 Likert-scale from strongly disagree (1) to strongly agree (7). 

The 5C measures attitude toward vaccination with 5 subscales: Calculation (“Engagement in extensive information searching”), Collective responsibility (“Inclination to protect others through herd immunity”), Complacency (“Perceived risks and threats of vaccine-preventable diseases and low desire for preventive action”), Confidence (“Trust in safety and effectiveness of vaccines”) and Constraints (“Affordability and accessibility barriers”) [47,51,52]. Due to the hypothetical nature of the COVID-19 scenario, we omitted the constraints statement, which we deemed more relevant during the implementation stage of vaccination [53]. Participants responded on a 7 Likert-scale from strongly disagree (1) to strongly agree (7).

### 2.5. Procedure

The survey took 6 min to complete on average [range 1.96–121.81]. All participants provided informed consent at the beginning of the survey and were randomly assigned to view one of the four NHS-like articles. To ensure validity of the survey, an attention check question followed the articles asking participants to select from three response options what the article had been about. Next, participants were asked to indicate their intention to vaccinate and attitude towards vaccination. To close, participants provided some demographic information (age, gender, ethnic group, employment and relationship status, number of children, level of education, household income, and residency). As part of the debrief participants were notified that the information provided in the NHS article, although scientifically accurate, had been modified for the purpose of the study. 

### 2.6. Ethical Approval

Ethical approval was obtained from the LSE Psychological and Behavioural Science Department and submitted [Ref:12236]. The study was pre-registered on an internal database placed on the Open Science Framework (10.17605/OSF.IO/Y9VUA). 

### 2.7. Statistical Analysis

Intention to vaccinate (Yes and No/Don’t know) was used as our primary outcome variable. As our secondary outcome variable, we first used a binomial logistic regression analysis to identify any demographic effects on vaccination intention. Next, we used chi-squared analyses of disease and risk frame on vaccination intention. Finally, we included disease and risk frame into the logistic regression model to estimate the effects of the intervention controlling for observed demographic differences between groups. 

Vaccination attitude was the second dependent variable in the experiment. We averaged the four facets of the VCI into one measure, and the four measured facets of the 5C (hereafter 4C) (Confidence, Complacency, Calculation and Community) into another. For both measures we first ran multinomial linear regression to check for demographic effects on vaccination attitude. Next, we used a 2 × 2 between-subjects ANOVA of risk and disease type to assess the effect of the intervention on VCI and 4C respectively. As with vaccination intention, we followed up by including the disease type variable and risk frame into the linear regression model to estimate the effects of intervention, whilst controlling for observed demographic differences between groups. 

By means of an exploratory analysis, we followed up on differences observed between white and non-white mothers using a 3-way binomial logistic regression analysis of disease type, risk frame, and ethnicity for vaccination intention.

## 3. Results 

### 3.1. Exclusion Criteria

Eight participants were excluded for failing to complete the survey in a given timeframe, i.e., between 2 and 20 min and one participant for selecting “outside the UK” as place of residence. No participants failed the attention check question. 

### 3.2. Demographic Characteristics of Participants

All participants (all mothers of at least one child under the age of 5) were female (93% between 25–44 years old; 85% white, 89% in a relationship). On average, mothers had 1.3 children under 5 (range = 1–4) and 87% lived in England. Participants were somewhat more highly educated (62% at least a Bachelor’s degree), working as much as (74%, 35% part-time and 8% self-employed), somewhat higher in income (61% above £40,000) and more likely to be in a relationship (89%) than the UK average (£29,900 household income, 75% working mothers, 43% mothers with degree, 14.7% parents not in a relationship) [54,55,56,57].

#### 3.2.1. In Relation to Vaccination Intention

Out of all participants (n = 542), 538 provided their vaccination intention. 399 intended to vaccinate or had vaccinated their child(ren) (Yes) and 139 did not or unsure (No/Don’t know). A binomial logistic regression of the above participant characteristics on child vaccination intention (DV) indicates that no characteristics significantly predicted vaccination intention, aside from ethnicity: non-white mothers reported a lower vaccination intention than white mothers in the sample (see Table 1 for details). The none-white group is made up of 36% black/African/Caribbean/black British, 25% Asian/Asian British, 21% mixed or multiple ethnic identities, 9% Other.

#### 3.2.2. In Relation to Vaccination Attitude

All participants (n = 542) provided their vaccination attitude (VCI: M = 18.3, SE = 0.147, 95% CI = 18.0–18.6); 4C: M = 45.9, SE = 0.411, 95% CI = 45.1–46.7). A linear regression of the participant characteristics on each of the attitude measures respectively indicates a consistent pattern. For both measures about 8% of the variance is explained by participant characteristics (VCI: F = 4.54, *p* < 0.001, Adjusted R^2^ = 0.0842; 4C: F = 4.39, *p* < 0.001; Adjusted R^2^ = 0.081). In both cases it was driven by ethnicity, number of children and household income (see Table 1 for details).

### 3.3. Treatment Effects 

#### 3.3.1. On Vaccination Intention

The chi-squared of type of risk by type of disease on vaccination intention indicates that parents were more likely to report vaccinating or wanting to vaccinate when the type of risk is to the child (79.63% of mothers) compared to the community (68.44% of mothers); [main effect of risk type: Z = 2.705, *p* = 0.007, OR = 2.135]. Parents were also more likely to vaccinate children against the Flu (84.3% of mothers) than against COVID-19 (63.6% of mothers; main effect of disease type [Z = 3.977, *p* < 0.001, OR = 3.271]. There was no interaction effect [Z = −0.551, *p* = 0.733] (see Figure 2 left panel).

#### 3.3.2. On Vaccination Attitude

Using a 2 × 2 between-subjects ANOVA of disease type and risk frame on VCI [F(3, 542) = 2.37, *p* = 0.070], we found a significant main effect of type of risk where those who received a frame emphasizing the risk to the child showed more positive attitude toward vaccination VCI [F(1, 542) = 4.339; *p* = 0.038, η^2^ = 0.008]. We found no main effect disease type [F (1, 542) = 2.374, *p* = 0.124, η^2^ = 0.004], nor an interaction effect [F(1, 542) = 0.002, *p* = 0.964, η^2^ < 0.001]. 

Despite similar directionality of results, the same 2 × 2 between-subjects ANOVA for 4C showed no significant effect [main effect risk frame: F(1, 542) = 3.82, *p* = 0.051, η^2^ = 0.007; main effect disease type: F(1, 542) = 3.62, *p* = 0.057, η^2^ = 0.007; interaction: F(1, 542) = 0.161, *p* = 0.689, η^2^ = 0.001]. 

(See Figure 3 middle and left panel for details and Appendix A for descriptives).

#### 3.3.3. Effect on Vaccination Intention & Attitude Controlling for Demographic Factors

To control for demographic factors, we added type of disease and risk frames into the binomial logistic regression on vaccination intention (DV: Yes vs. no/don’t know) and linear regressions for VCI and 4C. This confirms greater vaccination intention when risks to the child were highlighted [OR = 2.1, 95% CI = 1.23–3.70, *p* = 0.007], and a stronger vaccination intention against Flu than against COVID-19 [OR = 3.1, 95% CI = 1.82–5.866, *p* < 0.001], when controlling for demographic factors. Results remain largely unchanged (see Table 2).

### 3.4. Exploratory Analysis

As our study demonstrated consistent differences in vaccination intention and attitude for white and non-white mothers, we followed up on treatment effects for vaccination intention separately for these two groups (see Figure 4).

Using a binomial logistic regression with disease type, risk frame, ethnicity on vaccination intention [χ^2^ = 63.2, *p* < 0.001], we found a main effect of type of risk [Z = 3.272; *p* = 0.001, OR = 2.78], a main effect for disease type [Z = 3.49, *p* < 0.001, OR = 0.288], no main effect for ethnicity [Z = −0.932, *p* = 0.351, OR = 0.642], and an interaction effect between type of risk and ethnicity [Z = −2.06, *p* = 0.039, OR = 0.255], no interaction effect between type of disease and ethnicity [Z = 0.888, *p* = 0.374, OR = 2.286], and no interaction effect between all variables [Z = −0.566, *p* = 0.572, OR = 0.513]. In sum, white mothers showed a higher vaccination intention when given the child risk frame, whilst non-white mothers showed higher vaccination intention when given a community risk frame across both COVID-19 and Flu conditions (see Figure 4 for details).

## 4. Discussion

We predicted to observe differences between the two psychological frames on the vaccination attitude and the mothers’ intention to vaccinate their children for the two disease types. In particular, we expected that the ‘community frame’ would be more effective the lesser the threat to the child, posing that the child frame would be more effective in a flu context while the community frame would be more effective in a COVID-19 context. This is not entirely what we found. Our results showed that emphasizing risks of non-vaccination associated with the child’s health results in intention to vaccinate twice as high than when emphasizing risks to their community, for two types of disease contexts. We put forth a three-part explanation: first, a key barrier to vaccinating children is the perceived concerns about vaccine safety to the child’s health [58]. By emphasizing the risks of non-vaccination to the child’s health, parents who held this belief may have, at least temporarily, shifted their focus of their attention from the risk of vaccinating towards the risk of not vaccinating their child, increasing the intention to vaccinate [16].

Second, it is possible that mothers in our study may have underestimated the risks posed by both viruses to children under the age of 5 relative to the risks to the community. This is likely because the media emphasize the likelihood of health risks (greater in adult populations [NHS.UK]) than the size of the risk (significant in both adult and child populations) [24].

Third, and most consistent with literature prior to the COVID-19 pandemic, the social contract that binds parents to their children seems to have precedence over other social contracts in the community [25,31,59].

As a second contribution, our study provides evidence that emphasizing the risk for the child showed a demonstrably larger effect size in a COVID-19 context relative to the flu context in October 2020. We note that this was during a time where COVID-19 vaccinations were not yet on the market. Although the risks of severe illness posed to children by the common flu are greater than the ones posed by COVID-19, one possibility is that being in the middle of a pandemic, made COVID-19 highly salient in people’s minds in 2020 [20,60].

From our exploratory analyses, we also consistently find differences in vaccination intention and attitude between white compared to non-white mothers in our sample. We observed that more white mothers intended to vaccinate when exposed to a frame emphasizing the risk for the child than mothers who were non-white. These mothers seemed to respond better to the community frame. These exploratory findings may be worth pursuing with a larger sample size. We recognize that the exploratory analysis of racial differences is limited by the dichotomous distinction between white and non-white participants, due to its explorative nature. We suggest that follow up research looking at effects of the intervention across different ethnicities provides for a useful next step. There could be many reasons for the difference between groups, for one, the picture used in the Child articles, was displaying a white child with a white parent. Arguably, one hypothesis is that images of people from a different ethnicity might not have the same impact for non-white mothers [17]. Another could be that non-white population may be more community oriented [61]. A third may be to do with the NHS as the choice of messenger: ethnic minorities’ reportedly feel more rejected from public and political bodies such as the NHS, which is thought to contribute to a decrease of vaccination uptake and increased skepticism among those groups [62]. These findings highlight the importance of heterogeneous evaluations of risk in health [63] and heterogeneous behavioral design [64].

We also note several limitations to this experiment. First, at the time of the survey, COVID-19 vaccination hadn’t started in the UK, placing participants in front of a hypothetical scenario. Since the completion of the survey, much has changed in our understanding of COVID-19 variants and effects of vaccination amongst adults. This may explain a lower vaccination intention for COVID-19 relative to the common flu. A follow up may be able to observe the impact of more recent knowledge on COVID-19 vaccination intention. Nonetheless the risks of COVID-19 for children relative to the rest of the community, as well as the risks of vaccination have remained stable.

Second, it remains unclear whether highlighting the risk to the community has a positive impact on vaccination intention or whether it may have backfired due to free-riding motives [34,65], relative to a neutral frame. A follow up experiment could introduce a control group. Similarly, a follow up study would ideally measure behavioral outcomes, with regards to vaccination, to avoid the intention-behavior gap [16,37].

Third, we recognize that our sample consisted of participants who were—on average more highly educated than the general population of UK mothers, as these may share different attitudes toward scientific evidence in general [66]. To review the effectiveness of the intervention we compared vaccination attitude and intervention effects by education level and we found no significant differences.

Overall, our study supports that dealing with parental vaccine hesitancy benefits from an emphasis on the risk to the child rather than betting on their citizens’ altruistic behavior. Our findings add that this recommendation may apply to various vaccine-preventable childhood diseases, irrespective of the level of risk to children that are associated with the disease. In the low-risk case for COVID-19 and flu, we find vaccination intention to be twice as high when discussing risk to the child compared to risk to the community. Indeed, similar messages may demonstrate to be even more effective for diseases associated with higher dangers for children.

## 5. Conclusions

Alerting parents to the risks that non-vaccination poses to the children increases vaccination intention more than alerting parents to the risks to others, both in their community and across two vaccine-preventable diseases with low risks to the child. In line with previous studies, this suggests that vaccination campaigns benefit from emphasizing the risk to the child, without discarding the benefit that can come from emphasizing the risk to the community. Future research could aim to compare the effectiveness of both frames in a different cultural context where individualism is not as prevalent [67]. Future research could also consider testing the intervention effects with different caregiver compositions such as fathers or single parents.

## Figures and Tables

**Figure 1 vaccines-10-00720-f001:**
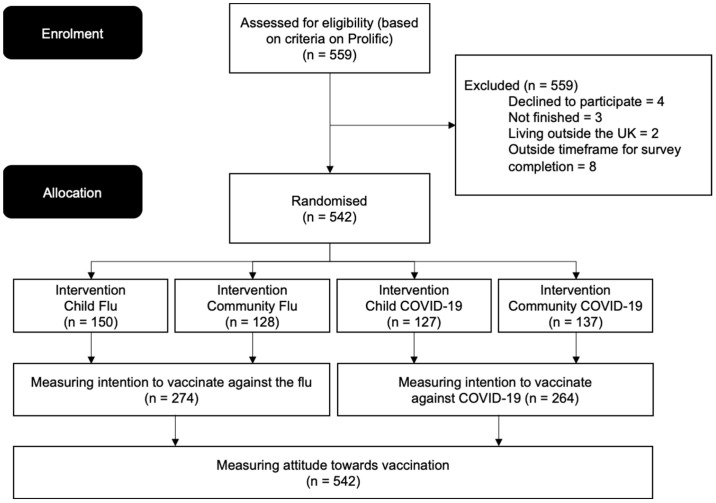
Consort flow diagram of experimental groups.

**Figure 2 vaccines-10-00720-f002:**
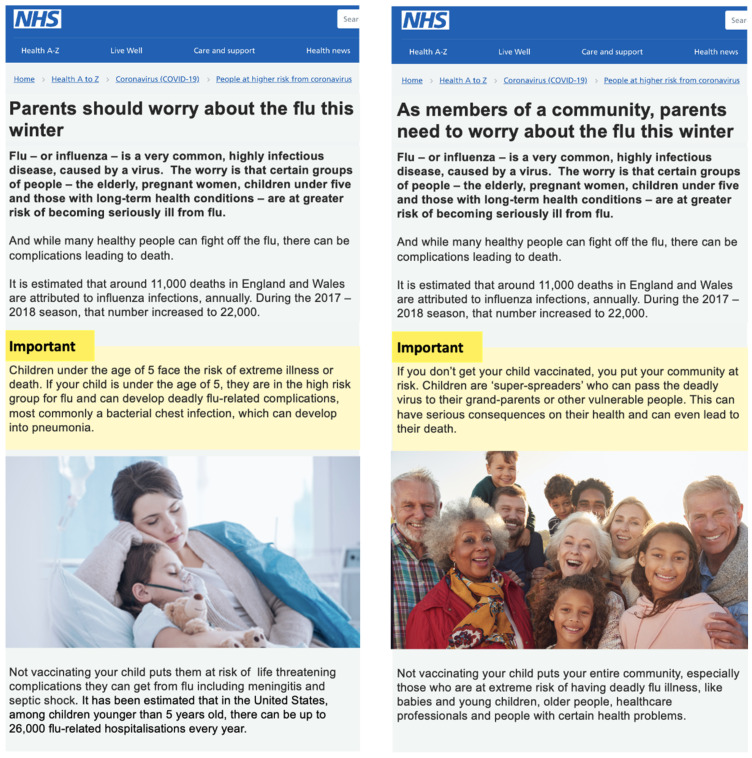
Schematic display of two of the NHS-like articles: flu with risks to child (**left**) and flu with risks to the community (**right**).

**Figure 3 vaccines-10-00720-f003:**
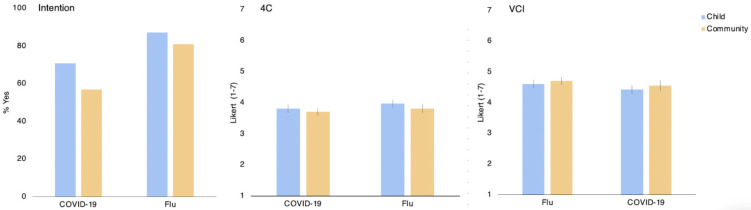
The effect of type of risk frame (child vs. community) by type of disease (Flu vs. COVID-19) discussed on vaccination intention (**left**) and attitude (4C **middle**, VCI **right**). Error bars display 95% Confidence Intervals.

**Figure 4 vaccines-10-00720-f004:**
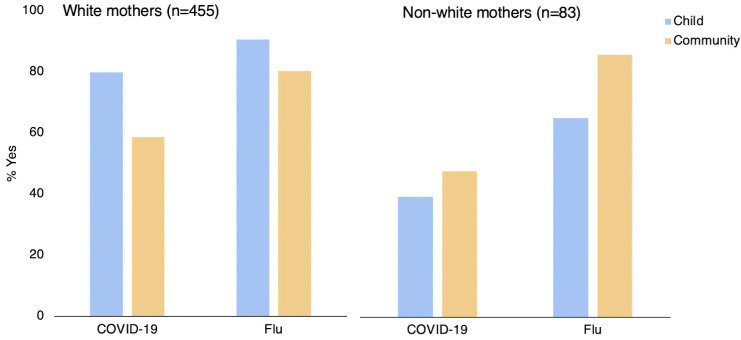
The effect of type of risk frame (child vs. community) by type of disease (Flu vs. COVID-19) on vaccination intention separated by white (**left**) and non-white (**right**) mothers.

**Table 1 vaccines-10-00720-t001:** Logistic of demographics on (i) vaccination intention (n = 538) and multinomial linear regression of demographics on (ii) Vaccine Confidence Index (VCI) and (iii) the 4C (Confidence, Complacency, Calculation and Community) attitude measures (n = 542).

Demographic Factors	Vaccination Intention			Demographic Factors	VCI			4C		
	OR	95% CI	*p*-Value		Stand. Est	95% CI	*p*-Value	Stand. Est	95% CI	*p*-Value
Ethnicity										
White (n = 455)	Reference			White (n = 459)	Reference			Reference		
Non-white (n = 83)	0.381	0.224–0.649	<0.001	Non-white (n = 83)	−0.440	−0.679–−0.202	<0.001	−0.457	−0.695–−0.218	<0.001
Relationship status										
In a relationship (n = 480)	Reference			In a relationship (n = 484)	Reference			Reference		
Not in a relationship (n = 58)	0.661	0.346–1.263	0.21	Not in a relationship (n = 58)	−0.237	−0.525–0.050	0.106	−0.162	−0.450–0.126	0.270
Education										
Higher education (n = 333)	Reference			Higher education (n = 335)	Reference			Reference		
Secondary education (n = 205)	0.821	0.535–1.259	0.391	Secondary education (n = 207)	0.005	−0.170–0.179	0.958	−0.106	−0.281–0.068	0.232
Region										
London (n = 60)	Reference			London (n = 61)	Reference			Reference		
England outside London (n = 408)	1.288	0.659–2.517	0.458	England outside London (n = 411)	0.156	−0.122–0.435	0.27	0.174	−0.105–0.453	0.220
Other UK (n = 68)	1.276	0.544–2.992	0.575	Other UK (n = 68)	0.190	−0.157–0.537	0.282	0.233	−0.115–0.580	0.189
Employment										
Employed Full-time (n = 164)	Reference			Employed Full-time (n = 166)	Reference			Reference		
Employed Part-time (n = 191)	1.012	0.600–1.709	0.963	Employed Part-time (n = 192)	0.067	−0.137–0.271	0.518	0.08	−0.124–0.285	0.440
Not working (n = 139)	0.845	0.486–1.469	0.551	Not working (n = 140)	0.078	−0.147–0.304	0.494	0.206	−0.002–0.432	0.074
Mother age										
18-24 (n = 34)	Reference			18-24 (n = 34)	Reference			Reference		
25-34 (n = 273)	1.211	0.529–2.769	0.651	25-34 (n = 276)	0.072	0.421–0.277	0.686	0.02	−0.329–0.370	0.909
35 or more (n = 231)	1.467	0.633–3.401	0.372	35 or more (n = 232)	0.038	−0.314–0.391	0.831	0.221	0.132–0.574	0.220
Number of children										
1 (n = 391)	Reference			1 (n = 395)	Reference			Reference		
2 or more (n = 147)	0.73	0.468–1.138	0.165	2 or more (n = 147)	−0.193	−0.379–−0.007	0.042	−0.241	−0.427–−0.055	0.011
Household income										
Below £30K (n = 149)	Reference			Below £30K (n = 151)	Reference			Reference		
From £30K to £50K (n = 193)	1.523	0.888–2.611	0.126	From £30K to £50K (n = 194)	0.416	0.191–0.642	<0.001	0.361	0.135–0.587	0.002
From £50K to £70K (n = 103)	1.506	0.782–2.898	0.22	From £50K to £70K (n = 104)	0.499	0.231–0.769	<0.001	0.474	0.205–0.744	<0.001
More than £70K (n = 93)	1.699	0.842–3.429	0.139	More than £70K (n = 93)	0.598	0.216–0.781	<0.001	0.398	0.115–0.681	0.006

**Table 2 vaccines-10-00720-t002:** Logistic of disease type and risk frame on (i) vaccination intention (n = 538) and multinomial linear regression of demographics on (ii) Vaccine Confidence Index (VCI) and (iii) the 4C (Confidence, Complacency, Calculation and Community) attitude measures (n = 542) controlling for person characteristics.

Intervention Effects and Demographic Variables	Vaccination Intention			Intervention Effects and Demographic Variables	VCI			4C		
	OR	95% CI	*p*-Value		Stand. Est	95% CI	*p*-Value	Stand. Est	95% CI	*p*-Value
Risk (IV1)										
Community frame (n = 263)	Reference			Community frame (n = 265)	Reference			Reference		
Child frame (n = 275)	2.135	1.232–3.698	0.007	Child frame (n = 277)	0.241	0.006–0.476	0.044	0.175	−0.060–0.411	0.143
Disease (IV2)										
COVID-19 (n =264)	Reference			COVID-19 (n = 264)	Reference			Reference		
Flu (n = 274)	3.271	1.824–5.866	<0.001	Flu (n = 278)	0.119	−0.113–0.352	0.314	0.139	−0.094–0.372	
Interaction										
IV1–IV2	0.783	0.327–1.871	0.582	IV1–IV2	−0.071	−0.398–0.255	0.666	−0.004	−0.331–0.323	0.982
Ethnicity										
White (n = 455)	Reference			White (n = 459)	Reference			Reference		
Non-white (n = 83)	0.368	0.210–0.645	<0.001	Non-white (n = 83)	−0.453	−0.453–−0.692	<0.001	−0.459	−0.684–−0.208	<0.001
Relationship status										
In a relationship (n = 480)	Reference			In a relationship (n = 484)	Reference			Reference		
Not in a relationship (n = 58)	0.624	0.315–1.238	0.177	Not in a relationship (n = 58)	−0.254	−0.542–0.033	0.083	−0.169	−0.458–0.119	0.249
Education										
Higher education (n = 333)	Reference			Higher education (n = 335)	Reference			Reference		
Secondary education (n = 205)	0.792	0.507–1.239	0.307	Secondary education (n = 207)	0.004	0.169–0.179	0.956	−0.108	−0.282–0.066	0.222
Region										
London (n = 60)	Reference			London (n = 61)	Reference			Reference		
England outside London (n = 408)	1.062	0.526–2.143	0.742	England outside London (n = 411)	0.124	−0.154–0.402	0.383	0.141	−0.138–0.419	0.321
Other UK (n = 68)	1.016	0.416–2.482	0.899	Other UK (n = 68)	0.150	−0.197–0.497	0.397	0.193	−0.155–0.541	0.277
Employment										
Employed Full-time (n = 164)	Reference			Employed Full-time (n = 166)	Reference			Reference		
Employed Part-time (n = 191)	1.077	0.624–1.857	0.790	Employed Part-time (n = 192)	0.078	−0.126–0.282	0.452	0.088	−0.117–0.292	0.399
Not working (n = 139)	0.933	0.525–1.659	0.815	Not working (n = 140)	0.088	−0.137–0.314	0.442	0.217	−0.009–0.443	0.060
Mother age										
18-24 (n = 34)	Reference			18-24 (n = 34)	Reference			Reference		
25-34 (n = 273)	1.156	0.564–2.234	0.739	25-34 (n = 276)	−0.087	−0.435–0.261	0.624	0.002	−0.347–0.351	0.990
35 or more (n = 231)	1.443	0.438–2.562	0.409	35 or more (n = 232)	0.025	−0.326–0.377	0.888	0.204	−0.149–0.556	0.257
Number of children										
1 (n = 391)	Reference			1 (n = 395)	Reference			Reference		
2 or more (n = 147)	0.738	0.463–1.176	0.202	2 or more (n = 147)	−0.172	−0.358–−0.001	0.069	−0.224	−0.410–−0.038	0.019
Household income										
Below £30K (n = 149)	Reference			Below £30K (n = 151)	Reference			Reference		
From £30K to £50K (n = 193)	1.659	0.943–2.919	0.079	From £30K to £50K (n = 194)	0.421	0.196–0.646	<0.001	0.368	0.143–0.594	0.001
From £50K to £70K (n = 103)	1.588	0.806–3.130	0.181	From £50K to £70K (n = 104)	0.500	0.232–0.768	<0.001	0.477	0.209–0.745	<0.001
More than £70K (n = 93)	1.725	0.830–3.586	0.144	More than £70K (n = 93)	0.504	0.223–0.786	<0.001	0.399	0.117–0.682	0.006

## Data Availability

Data is contained within Appendix A. The data presented in this study are available in the Appendix A.

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
