# Peer review of "An Online Experiment of NHS Information Framing on Mothers’ Vaccination Intention of Children against COVID-19"

_vaccines, 2022, doi:10.3390/vaccines10050720_

Round 1

Reviewer 1 Report

The article "An online experiment of NHS information framing on mother’s vaccination intentions of children against COVID-19" is certainly interesting, well thought out, and well written. The research question is useful both in terms of deepening the scientific understanding of complex phenomena and for planning public health interventions. The study design seems appropriate for testing the study hypothesis.
The article would be publishable as it is, but I believe that the article could be further improved and therefore I have a few questions and suggestions for the authors.
In mentioning the use of Prolific Academic could the authors explain how they used the website and give some formal references?
Could the authors loosen up the paragraph on limitations, making it easier to understand, including a reference to possible bias due to the use of a network of people with a particular attitude towards science, compared to people with a low or different attitude?
In the conclusions, couldn't the authors put forward some hypothesis, at least on the research level, to enhance in the future also the community context, which comes out so frustrated compared to the individual context?
In the two tables in the supplementary material, a legend explaining the acronyms would avoid searching through the text.

Reviewer 2 Report

The authors have meticulously investigated the reason for hesitancy on COVID-19 vaccines in children under 5 years. The study is well-designed and supports the main findings of the study. Few points to be pointed out, most of which I think that cannot be solved in the current data set.

1. One thing is that, even though the decision to vaccine their child may majorly depend on the mother of the child, the role of fathers could have been investigated. The decision to make vaccine against COVID-19 maybe somewhat different from previous vaccine preventable diseases (such as flu). There may also be single care givers; in this case, the study result could be biased due to selection.

2. Even though the explanation on different attitudes toward vaccination between white and non-whites is definitely reasonable, however, this grouping is somewhat dichotomous. As each ethnic group may have their own reason towards or against vaccination, the comparisons could have been made in a multiple manner.

3. Furthermore, the authors could have looked into the conditions of the child of the respondents. Childs with underlying disease will benefit much more than those with no underlying disease with COVID-19 vaccination. Therefore, the attitude towards vaccination may vary from this. In this case, parents with underlying conditions may have different attitude towards vaccination. It is not sure whether the main findings from the study would be intensified in the patients of child with medical conditions; they may think or act different, against expectations.

Above all, I believe that this study adds substantial knowledge to the literature as the study is based on the assumption that all the parents to be reasonable/rational individuals.
